# Protist Diversity and Metabolic Strategy in Freshwater Lakes Are Shaped by Trophic State and Watershed Land Use on a Continental Scale

Rebecca E. Garner,[a,f] Susanne A. Kraemer,[a,b,c,f] Vera E. Onana,[a,f] Yannick Huot,[d,f] Irene Gregory-Eaves,[e,f] David A. Walsh[a,f]

aDepartment of Biology, Concordia University, Montreal, Quebec, Canada
bGenome Centre, Department of Microbiology & Immunology, McGill University, Montreal, Quebec, Canada
cDepartment of Civil Engineering, McGill University, Montreal, Quebec, Canada
dDépartement de géomatique appliquée, Université de Sherbrooke, Sherbrooke, Quebec, Canada
eDepartment of Biology, McGill University, Montreal, Quebec, Canada
fGroupe de recherche interuniversitaire en limnologie, Montreal, Quebec, Canada

**ABSTRACT** Protists play key roles in aquatic food webs as primary producers, predators, nutrient recyclers, and symbionts. However, a comprehensive view of protist diversity in freshwaters has been challenged by the immense environmental heterogeneity among lakes worldwide. We assessed protist diversity in the surface waters of 366 freshwater lakes across a north temperate to subarctic range covering nearly 8.4 million $km^2$ of Canada. Sampled lakes represented broad gradients in size, trophic state, and watershed land use. Hypereutrophic lakes contained the least diverse and most distinct protist communities relative to nutrient-poor lakes. Greater taxonomic variation among eutrophic lakes was mainly a product of heterotroph and mixotroph diversity, whereas phototroph assemblages were more similar under high-nutrient conditions. Overall, local physicochemical factors, particularly ion and nutrient concentrations, elicited the strongest responses in community structure, far outweighing the effects of geographic gradients. Despite their contrasting distribution patterns, obligate phototroph and heterotroph turnover was predicted by an overlapping set of environmental factors, while the metabolic plasticity of mixotrophs may have made them less predictable. Notably, protist diversity was associated with variation in watershed soil pH and agricultural crop coverage, pointing to human impact on the land-water interface that has not been previously identified in studies on smaller scales. Our study exposes the importance of both within-lake and external watershed characteristics in explaining protist diversity and biogeography, critical information for further developing an understanding of how freshwater lakes and their watersheds are impacted by anthropogenic stressors.

**IMPORTANCE** Freshwater lakes are experiencing rapid changes under accelerated anthropogenic stress and a warming climate. Microorganisms underpin aquatic food webs, yet little is known about how freshwater microbial communities are responding to human impact. Here, we assessed the diversity of protists and their myriad ecological roles in lakes varying in size across watersheds experiencing a range of land use pressures by leveraging data from a continental-scale survey of Canadian lakes. We found evidence of human impact on protist assemblages through an association with lake trophic state and extending to agricultural activity and soil characteristics in the surrounding watershed. Furthermore, trophic state appeared to explain the distributions of phototrophic and heterotrophic protists in contrasting ways. Our findings highlight the vulnerability of lake ecosystems to increased land use and the importance of assessing terrestrial interfaces to elucidate freshwater ecosystem dynamics.

Address correspondence to Rebecca E. Garner, rebecca.garner@mail.concordia.ca, or David A. Walsh, david.walsh@concordia.ca.

The authors declare no conflict of interest.

**KEYWORDS** microbial eukaryotes, trophic state, plankton, phototrophy, heterotrophy, mixotrophy, human impact

Protists have evolved a vast morphological and ecological diversity (1). In aquatic ecosystems, protists play key roles in the transfer of energy and nutrients by converting sunlight into chemical energy, remineralizing organic matter, controlling microbial biomass, feeding higher trophic levels, and maintaining symbioses, some of which recruit prokaryotic metabolisms (2–4).

Elucidating protist diversity in lakes is relevant for clarifying microbial distributions across a wide array of environmental conditions and for tracking the health of critical freshwaters. Covering <1% of Earth's surface (5), lakes contribute disproportionately to the global carbon cycle (6–8) and hold essential water resources (9). Lakes display a rich environmental heterogeneity, generated by integrating fluxes of materials and energy from their catchments and airsheds (10, 11). Freshwater lakes are hotspots of biodiversity, collectively containing higher levels of eukaryote richness and endemism (12) and protist community turnover (13) than the marine and terrestrial realms. In the Anthropocene, lakes are increasingly altered by eutrophication (14), warming temperatures (15), deoxygenation (16), salinization (17), and myriad other persistent and emergent stressors (18). There is accumulating evidence that anthropogenic modifications to lake habitats affect protist assemblages (19–21), in turn influencing ecosystem dynamics.

Efforts to study protists from a variety of biomes have unearthed a vast diversity and begun to map their distributions on a global scale (22–24). Such investigations have provided insights into protist environmental preferences and community assembly processes (25–28), food web dynamics (29–33), symbioses (34–36), viruses (37), functional traits (38), and bioindicator values (39, 40). Large-scale lake surveys have shown that protist assemblages are shaped by broad biogeographic patterns (41–43) but are also influenced by local environmental factors and interactions with bacteria (44). Phytoplankton surveys recapitulate these observations while exclusively investigating photosynthetic taxa including *Cyanobacteria* (45–47). However, there is no clear understanding yet of the distributions of different trophic life history strategies and the environmental drivers underlying their diversity. This knowledge gap is particularly glaring, since heterotrophy is likely the most abundant trophic mode (48). Meanwhile, mixotrophs have dramatically altered our view of plankton food webs by combining primary production and prey consumption (49), sometimes surpassing obligate heterotrophs as the leading grazers of bacteria in lakes (50, 51).

In this study, 18S rRNA gene amplicon sequencing was used to investigate the distributions of protists in the surface waters of 366 freshwater lakes across a north temperate to subarctic continental range. We hypothesized that (i) protist diversity at the local scale and community turnover decrease under high-nutrient conditions; (ii) trophic groups respond to different environmental factors, and specifically, phototrophs are more sensitive than heterotrophs to bottom-up resource availability; and (iii) because lakes integrate their catchments, protist diversity in lakes should reflect watershed conditions. This project was conducted within the LakePulse survey, which sampled hundreds of lakes of different sizes in watersheds across a gradient of human impact with the primary aim to assess lake health through a multidisciplinary lens (52). The current study fills a gap in the mapping of microbial biogeography through this first standardized assessment of protist diversity across Canada, which stewards the greatest abundance of lakes worldwide (53). Our study draws attention to the diversity of protists and the ecological patterns that emerge in a broad collection of newly explored habitats being reshaped by increasing human impact.

## RESULTS

**Sampled lakes and watersheds display high environmental heterogeneity.** Protist assemblages were surveyed in the euphotic zones of 366 freshwater to oligosaline lakes in 12 ecozones across Canada (43 to 68°N, 53 to 141°W) (Fig. 1). Watersheds

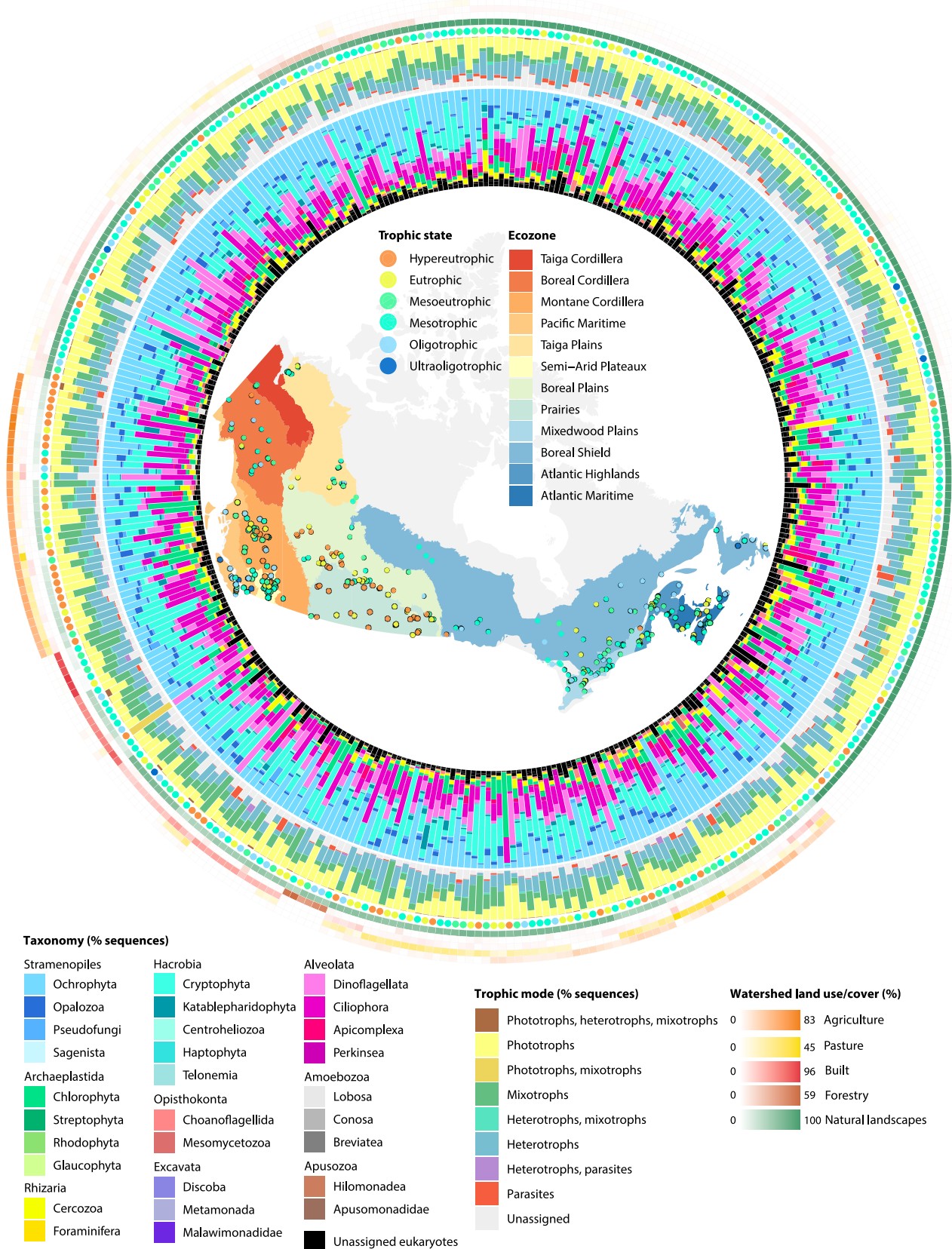

**FIG 1** Diversity and distributions of protists across 366 Canadian lakes. Lake trophic states and ecozones are shown on the map of sampling sites as coloured circles and polygons, respectively. The relative sequence abundance of protist taxonomic divisions in each lake is represented in the inner

ranged widely in area (0.3 to 9,332.3 km²) and were characterized by a variety of human population densities (0 to 3,785 people/km²) and land use, including different proportions of crop agriculture (0 to 81%) and built development (0 to 93%). Lakes had a wide range of surface areas (0.05 to 99.66 km²) and maximum depths (1 to >150 m) and were either vertically mixed (~40% of lakes) or thermally stratified (~60%) at the time of sampling. Physicochemical conditions differed substantially across lakes, as represented by a broad pH gradient (5.6 to 10.2) and ultraoligotrophic to hypereutrophic states evaluated by total phosphorus (TP) concentrations (2 to 2,484 $\mu$g/L) (Fig. 1).

Sampled lakes and watersheds reflected regional variation in environmental conditions, including anthropogenic gradients (see Fig. S1 in the supplemental material). Lakes in northern Canada (Taiga Cordillera, Boreal Cordillera, and Taiga Plains ecozones) were subject to the coldest climates and lowest intensities and proportions of land use within their watersheds. Lakes in western Canada (Montane Cordillera, Pacific Maritime, and Semi-Arid Plateaux) were the deepest on average and located in watersheds with the largest proportions of harvested forests. In central Canada (Boreal Plains and Prairies), a region dominated by plains and prairies with extensive agriculture, lakes were generally shallow, exposed to winds, productive, and high in pH, carbon, ions, and nutrients. Lakes in eastern Canada (Mixedwood Plains, Boreal Shield, Atlantic Highlands, and Atlantic Maritime) had the warmest surface waters and generally the most built-up landscapes, a feature shared with watersheds in the Pacific Maritime. Overall, nutrient-, ion-, and carbon-rich waters, and high-pH lake conditions were most often observed in agricultural watersheds with alkaline soils (Fig. S2).

**Lakes support taxonomically and functionally diverse protist assemblages.** Eukaryotic diversity was assessed through the sequencing of 18S rRNA gene fragments amplified from DNA collected in 0.22- to 100-$\mu$m surface water particles. A total of 15,848 amplicon sequence variants (ASVs) were inferred in 17,749,930 sequences across 366 lake samples. A final data set of 13,046 putative protist ASVs encompassing 14,622,273 sequences was retained after ASVs assigned to animals, fungi, and plants were removed (Table S1). The rarefaction of pooled samples showed that the sampling of new ASVs plateaued toward 2,000,000 sequences, signaling that the global sequencing effort had exhaustively captured the protist diversity targeted by the primer pair (Fig. S3). The most abundant taxonomic groups were Ochrophyta (24% of all sequences), Cryptophyta (18%), Ciliophora (15%), and Dinoflagellata (11%) (Fig. 1). Lineages with the highest ASV richness were Ochrophyta (22% of all ASVs), Dinoflagellata (12%), and Chlorophyta (11%).

We analyzed ASV incidence to assess the contribution of individual assemblages to total landscape diversity. New genotypes accumulated at a high rate in the first ~100 randomly ordered assemblages, followed by a gradual deceleration (Fig. 2A). The majority of ASVs were restricted to one or a few lakes (Fig. 2B). A smaller number of ASVs were distributed widely, including one ASV assigned to *Cryptomonas curvata* (Cryptophyta) that was ubiquitous yet highly variable in relative abundance (0.0031 to 64%) across all the lakes sampled.

Trophic functions were assigned to ASVs representing 85% of the total sequence space by leveraging natural history descriptions summarized from the literature (Fig. 1; Table S1). Of the functionally annotated ASVs, most were classified as obligate phototrophs (32% of all sequences). The most abundant phototrophs were classified as Chrysophyceae (Ochrophyta; 16% of all sequences), Bacillariophyta (Ochrophyta; 3%), and Chlorophyceae (Chlorophyta; 5%). Bacterivory (20% of all sequences) and mixotrophy (20%) were the next most abundant trophic modes. The most abundant mixotrophs were classified as Cryptophyceae (Cryptophyta; 17% of all sequences) and Dinophyceae (Dinoflagellata; 2%). Cytotrophy (i.e., feeding on other protists), parasitism, commensalism,

**FIG 1** Legend (Continued)

tract of bar plots. The middle tract of bar plots shows the relative sequence abundance of trophic modes. The heat map on the outer edge illustrates the proportions of land use and land cover associated with the watershed of each lake, whose trophic state is represented in an adjacent coloured circle. Watershed land use proportions are hierarchically clustered to highlight the relationship between agriculture and trophic state.

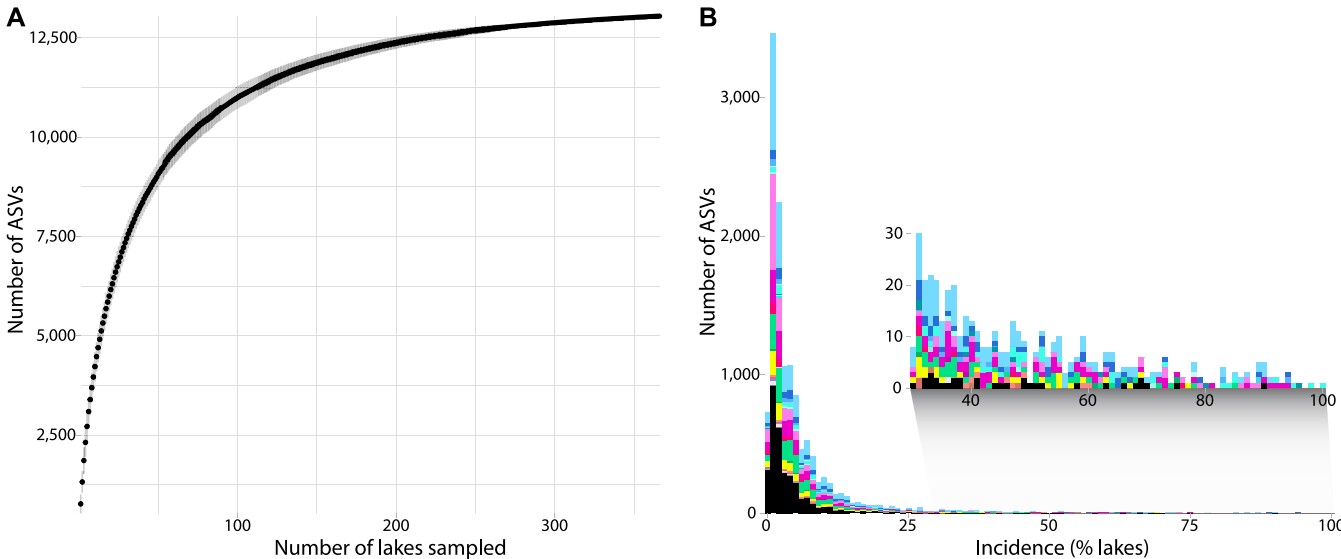

**FIG 2** Accumulation and incidence of genotypes across lakes. (A) Accumulation curve of genotypes in a random ordering of lakes. Vertical bars are standard deviations. The accumulation of new genotypes was rapid in the first ~100 lakes, followed by a gradual deceleration. (B) Incidence of genotypes across lakes. Most taxa are distributed across one or a few lakes, whereas a few taxa (magnified in the inset) are widely distributed. ASV taxonomic classifications are coloured according to the taxonomic divisions in Fig. 1.

saprotrophy, and osmotrophy were detected to a lesser extent. Parasites (3% of all sequences) were most abundant in the Coccidiomorphea (Apicomplexa; 2%) and Oomycota (Pseudofungi; 1%). Heterotrophs—which broadly encompassed bacterivores, cytotrophs, saprotrophs, and osmotrophs—accounted for 27% of all sequences. Heterotrophy was most abundant in the ciliates Spirotrichea (9% of all sequences), Oligohymenophorea (2%), and Litostomatea (2%) and in other lineages, including Bicoecea (Opalozoa; 3%) and Katablepharidaceae (Katablepharidophyta; 2%).

Genotypic similarity to known protist 18S rRNA gene diversity was assessed through the global alignment of ASVs to V7 region fragments in the Protist Ribosomal Reference database (PR²) (Fig. S4). Of the 12,511 ASVs that met the threshold global alignment length, the majority showed high sequence similarity with references in the database. Most ASVs (58%) were ≥96% identical to references. ASVs with 100% similarity to reference sequences occupied 56% of the total sequence space, while ASVs with ≥96% similarity occupied 91% of sequences. The most novel genotypes included the 738 ASVs with <90% sequence similarity to PR² references and were either not assigned to a supergroup (415 ASVs) or primarily classified as Opisthokonta (141 ASVs), Alveolata (88 ASVs), or Stramenopiles (30 ASVs).

Local diversity at each lake was estimated by richness (67 to 1,275 ASVs) (Fig. S5), the Shannon index (0.17 to 5.67) (Fig. 3A), Pielou's evenness index (0.04 to 0.83), and Faith's phylogenetic diversity index (8.15 to 76.04). The Shannon index was negatively correlated with magnesium ($r = -0.33$), total nitrogen (TN; $r = -0.32$), dissolved inorganic carbon (DIC; $r = -0.31$), and potassium ($r = -0.30$) concentrations (all correlations had $P$ values of <0.001). Analyses of variance (ANOVAs) followed by *post hoc* Tukey's tests comparing the association of trophic state with local diversity showed that mean richness, Shannon diversity, and evenness were significantly lower in hypereutrophic lakes than in eutrophic, mesoeutrophic, mesotrophic, or oligotrophic lakes (all $P \leq 0.002$) (Fig. 3B).

**Protist assemblages vary regionally and across lake trophic states.** Next, we looked at how communities varied in taxonomic and phylogenetic composition among lakes. A principal component analysis (PCA) of ASV assemblages showed a clear pattern of taxonomic variation by lake trophic state and ecozone along the first dimension (Fig. 4A). Assemblages in the typically nutrient- and ion-rich lakes of the Prairies and Boreal Plains were distinguished from assemblages in the lower-nutrient lakes of the Boreal Shield and other eastern regions. Cryptophyte diversity (*Cryptomonas*, *Geminigera*, *Plagioselmis*, and

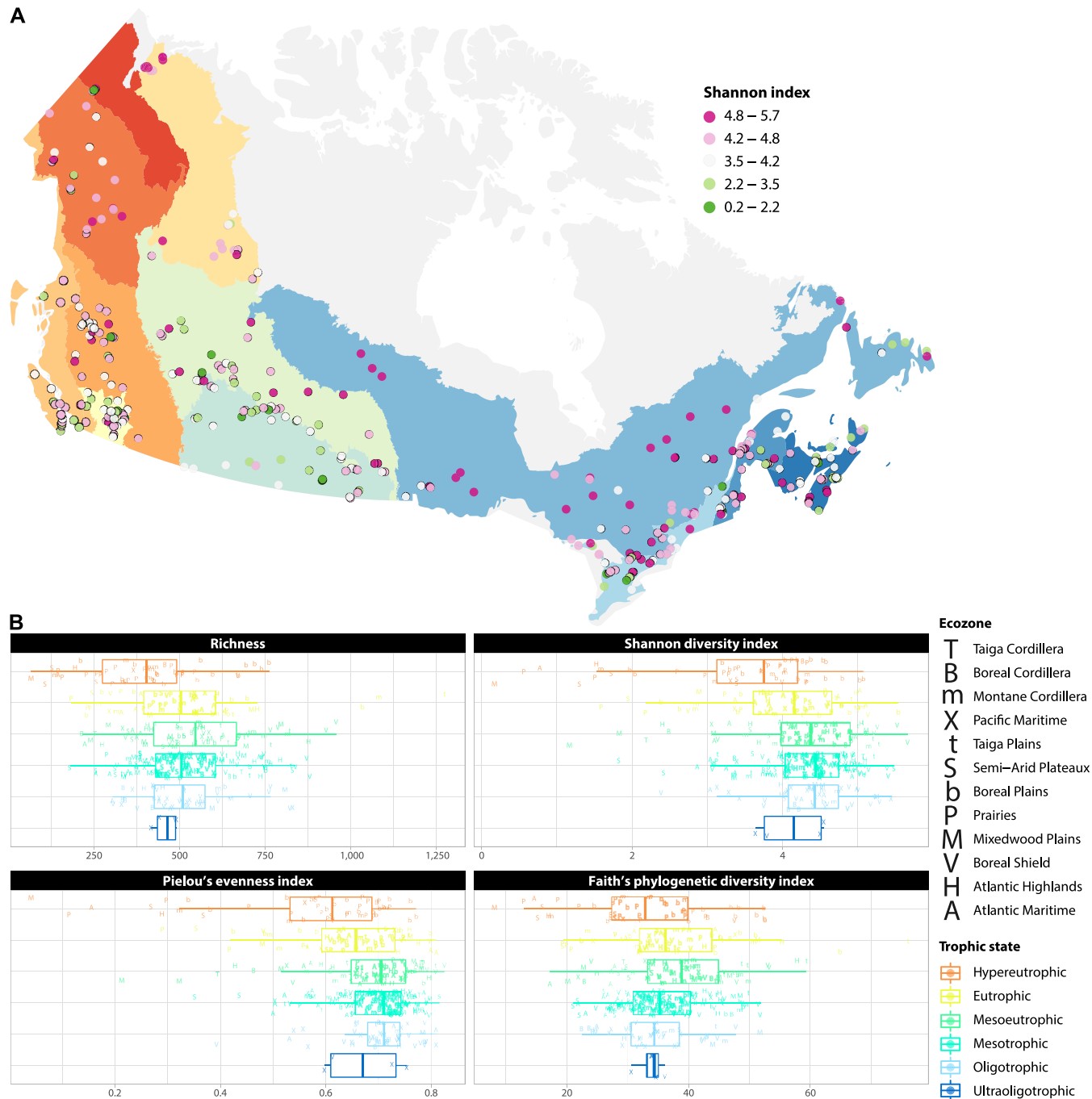

**FIG 3** Local protist diversity of each lake. (A) Rarefied Shannon diversity index calculated for each protist assemblage across the Canadian landscape. Ecozones are identified in the key of Fig. 1. (B) Local diversity metrics categorized by lake trophic state. TP concentration thresholds and sample sizes of lakes categorized under each trophic state are as follows: ultraoligotrophic (TP concentration, $<4$ $\mu$g/L; $n = 4$), oligotrophic (4 to 10 $\mu$g/L; $n = 48$), mesotrophic (10 to 20 $\mu$g/L; $n = 132$), mesoeutrophic (20 to 35 $\mu$g/L; $n = 71$), eutrophic (35 to 100 $\mu$g/L; $n = 61$), and hypereutrophic ($>100$ $\mu$g/L; $n = 50$).

*Komma* species) contributed the most strongly to the variation among assemblages (Fig. 4A). A principal coordinate analysis (PCoA) of generalized UniFrac distances between assemblages showed patterns of phylogenetic variation that were highly congruent with the taxonomic variation observed in the PCA, as evaluated by an RV coefficient correlating the first two dimensions of each ordination (RV = 67%, $P = 1.3 \times 10^{-98}$) (Fig. 4B).

Taxonomically distinct assemblages as quantified by local contributions to $\beta$-diversity (LCBD) were mostly localized in the Prairies and Boreal Plains, with a few high-LCBD

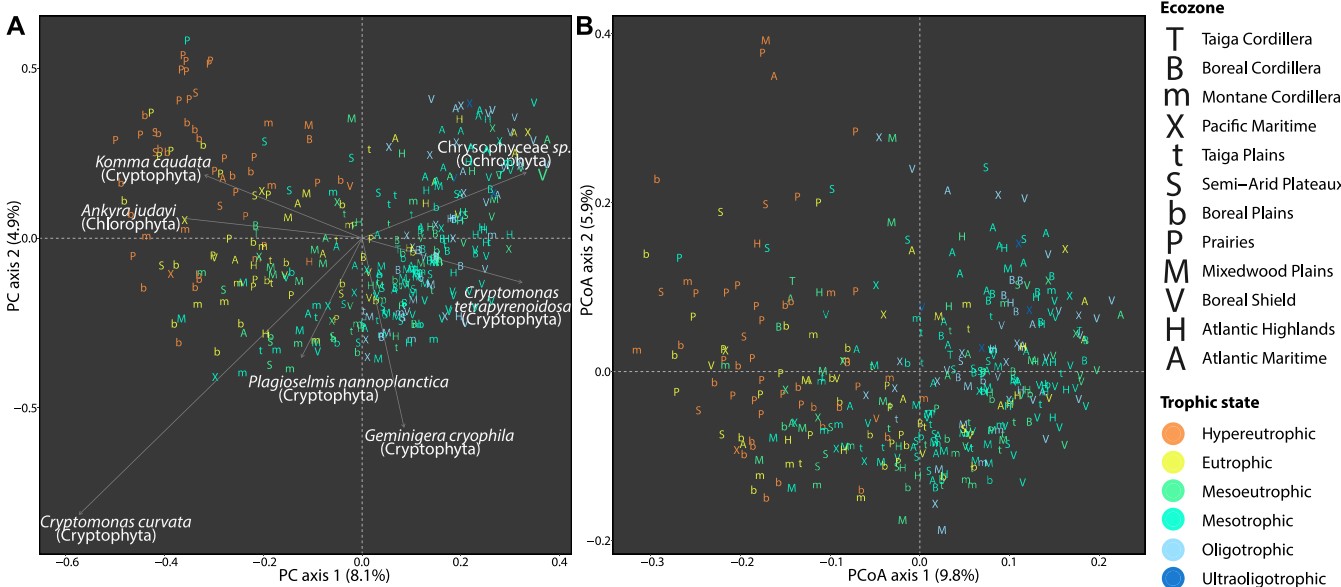

**FIG 4** Taxonomic and phylogenetic variation of protist assemblages among lakes. (A) PCA of the taxonomic variation among protist assemblages at the level of individual ASVs. The relative contributions and taxonomic assignments are shown for the top 7 ASVs (an arbitrary cutoff selected for illustrative clarity) contributing to the variation explained by the first two principal component dimensions. (B) PCoA of the phylogenetic variation among protist assemblages.

assemblages scattered across other regions (Fig. 5). Beta-diversity partitioning showed that the taxonomic dissimilarities between assemblages were primarily generated through ASV turnover (75% of total variance) and to a lesser extent through differences in richness (25%). Highly significant ($P < 0.001$) positive correlations were detected between LCBD and a complement of physicochemical variables, including potassium

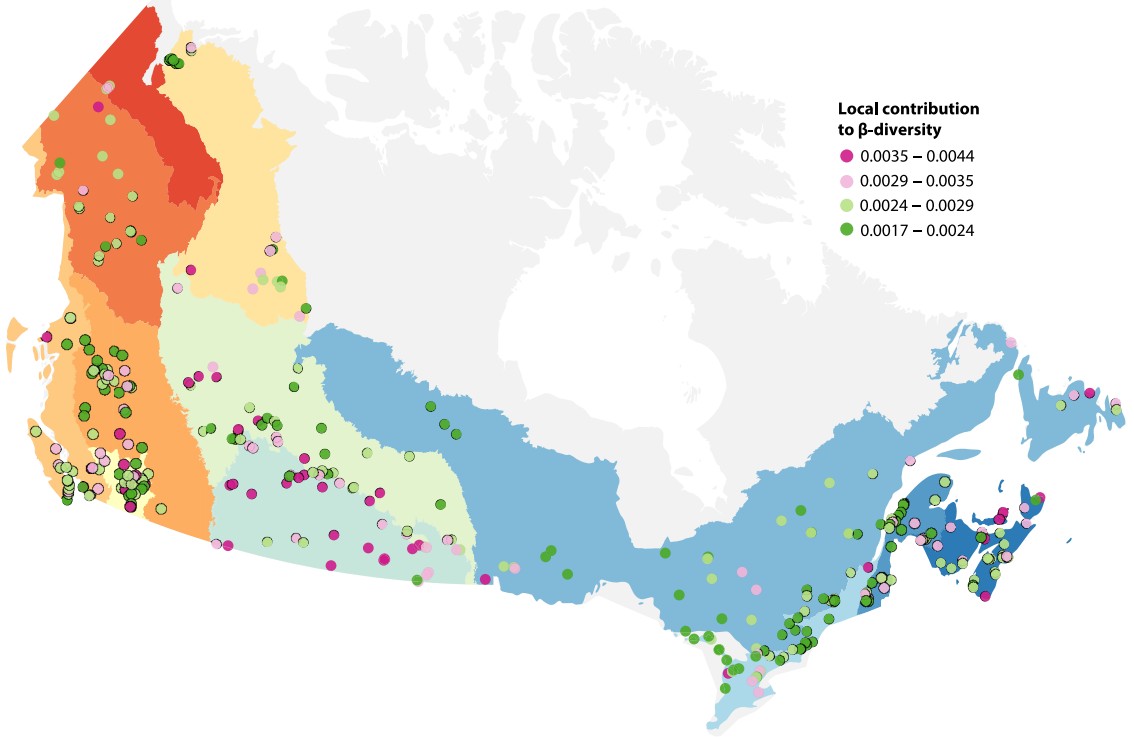

**FIG 5** Local contribution to $\beta$-diversity (LCBD) of protist assemblages across the Canadian landscape. LCBD describes the taxonomic uniqueness of a given assemblage, i.e., how much the taxonomic composition differs from the rest of the communities in the landscape. Ecozones are identified in the map legend of Fig. 1.

**TABLE 1** Percent deviance explained by generalized dissimilarity models (GDMs) fitting the responses of protist assemblages to environmental gradients[a]

| | Deviance explained (%) | | | | |
| | All protists | | Trophic mode[b] | | |
| Explanatory variables | Taxonomy | Phylogeny | Phototrophs | Heterotrophs | Mixotrophs |
|---|---|---|---|---|---|
| Physicochemistry | 38 | 33 | 35 | 42 | 18 |
| Watershed | 15 | 16 | 20 | 19 | NS |
| Morphometry | 6 | 8 | 5 | NS | NS |
| Weather | NS | NS | NS | 3 | NS |
| Geography | NS | NS | NS | 2 | NS |

[a]GDMs were constructed using various community response data and categories of environmental explanatory variables. NS, model was not statistically significant ($P \geq 0.05$).
[b]Analysis was performed only on taxonomic composition response data.

($r = 0.53$), TN ($r = 0.50$), DIC ($r = 0.45$), TP ($r = 0.44$), dissolved organic carbon (DOC; $r = 0.42$), magnesium ($r = 0.40$), sodium ($r = 0.38$), sulfate ($r = 0.37$), chlorophyll $a$ ($r = 0.33$), lake colour ($r = 0.22$), and pH ($r = 0.22$). LCBD further correlated ($P < 0.001$) with watershed crop cover ($r = 0.35$) and soil pH ($r = 0.27$) and correlated negatively with soil nitrogen ($r = -0.25$). LCBD was negatively correlated with local diversity, estimated as ASV richness ($r = -0.44$) and the Shannon ($r = -0.54$) and Faith's phylogenetic diversity ($r = -0.34$) indices (all $P < 0.001$).

**Physicochemical and watershed conditions predict community turnover.** Following the previous observations of high community variability across an environmentally heterogeneous set of lakes, we evaluated the drivers of taxonomic and phylogenetic turnover based on five categories of environmental variables: (i) geography (i.e., latitude, longitude, and altitude), (ii) weather, (iii) lake morphometry, (iv) physicochemistry, and (v) watershed characteristics, including land use and edaphic properties.

We employed generalized dissimilarity models (GDMs) to detect nonlinear trends between community turnover and environmental gradients. Physicochemical factors explained the highest deviance in GDMs modeling taxonomic or phylogenetic turnover. In order of decreasing strength, DIC, TP, chlorophyll $a$, magnesium, pH, potassium, lake colour, surface temperature, and DOC were statistically significant predictors of taxonomic turnover, whereas chlorophyll $a$, magnesium, TN, pH, calcium, and colour were significant predictors of phylogenetic turnover. After physicochemistry, watershed characteristics were the most important predictors of community turnover, with both taxonomic and phylogenetic diversity responding most strongly to soil pH and then proportion of cropland cover. The volumetric fraction of soil coarse fragments and natural landscape coverage were additional predictors of taxonomic and phylogenetic turnover, respectively. Lake morphometry, comprising maximum depth, watershed slope, and shoreline circularity, had relatively weak effects on turnover. Weather and geography did not generate statistically significant GDMs. Model deviances are summarized in Table 1, and the partial effects of individual variables are summarized in Fig. S6 and Table S2.

**Trophic strategies exhibit contrasting distributions.** To examine the distributions of different trophic strategies, we investigated the taxonomic variation of phototroph, heterotroph, and mixotroph communities in separate PCAs (Fig. 6). Reflecting the taxonomic variation emerging at the whole-community level, assemblages of each trophic mode were distinguished by lake trophic state and ecozone along the first dimension. However, phototrophs displayed taxonomic variation patterns contrasting those of heterotrophs and mixotrophs. To compare the taxonomic turnover of phototrophs, heterotrophs, or mixotrophs among lakes of the same trophic state, we measured the distances of assemblages to the trophic state median within the two-dimensional principal coordinate space. The mean distance of phototroph assemblages to the trophic state median (i.e., turnover) was significantly lower in hypereutrophic lakes than in eutrophic, mesoeutrophic, mesotrophic, or oligotrophic lakes (all $P < 0.001$) (Fig. S7). In contrast, the mean distance of heterotroph assemblages was significantly higher in

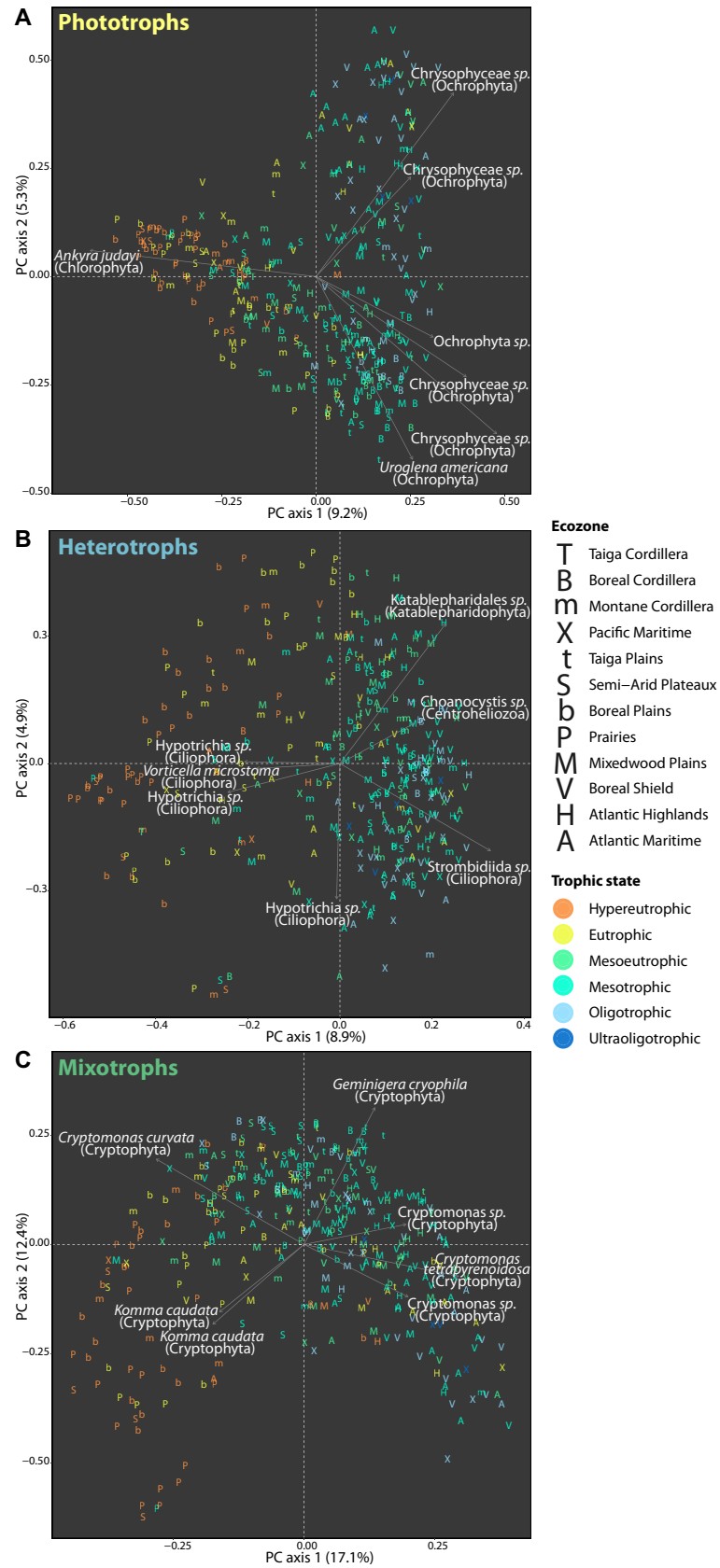

**FIG 6** PCA of the taxonomic variation among (A) phototroph, (B) heterotroph, and (C) mixotroph assemblages at the level of individual ASVs. The relative contributions and taxonomic assignments

eutrophic lakes than in mesotrophic or oligotrophic lakes (all $P \leq 0.007$) (Fig. S7). Compared with either phototroph or heterotroph assemblages, mixotroph assemblages were highly dispersed within trophic state groups (Fig. S7).

Partitioning protist assemblages by trophic function further allowed us to examine the responses of different groups to environmental conditions. GDMs showed that phototrophs, heterotrophs, and mixotrophs each responded most strongly to physicochemical gradients. Chlorophyll $a$, DIC, potassium, and pH were predictors common to the three trophic modes, while calcium, colour, and chloride were additional predictors of phototroph turnover, and TP, surface temperature, sulfate, TN, colour, and chloride were additional predictors of heterotroph turnover. Chlorophyll $a$ followed by potassium were the top predictors of phototroph turnover, while DIC followed by chlorophyll $a$ were the top predictors of heterotroph and mixotroph turnover. Watershed characteristics explained an important amount of deviance in phototroph and heterotroph turnover: phototroph turnover was explained by soil pH, crop coverage, soil organic carbon density, and built development, whereas heterotroph turnover was additionally explained by soil coarse fragments. Only phototroph turnover was predicted by lake morphometry variables (watershed slope, lake maximum depth, and circularity). Heterotroph turnover was weakly explained by weather, specifically air temperature and wind speed, and geography, specifically, altitude and distances between lakes. Mixotroph turnover was the least predictable from environmental conditions, as physicochemical factors explained less deviance than for obligate phototroph or heterotroph turnover and GDMs modeled on geography, weather, lake morphometry, and watershed characteristics were not statistically significant. Model deviances are summarized in Table 1, and the partial effects of individual variables are summarized in Fig. S8 and Table S2.

## DISCUSSION

**Protist diversity unveiled at a continental scale.** Establishing a comprehensive perspective of freshwater protist diversity is challenging given the substantial environmental heterogeneity of the millions of lakes distributed globally. Our study fills a sizable gap by mapping protist distributions across hundreds of lakes spanning an area covering nearly 8.4 million km² in the largest study of its kind to employ a standardized sampling scheme (52). Protist diversity was determined in the sunlit surface waters of 366 freshwater lakes varying in size and degree of human impact on the watershed. The sampled biomass bridged cell diameters across 4 orders of magnitude (0.22 to 100 $\mu$m), allowing us to recover pico- to microscale organisms, many of which are not resolved under standard light microscopy and which constitute an expansive microbial diversity encompassing the major eukaryotic lineages and trophic strategies. Our global rarefaction analysis showed that the sequencing effort provided a reasonable estimate of the genotypic variation captured by the primer pair. The rapid accumulation of genotypes across sites indicated that sampling hundreds of lakes was required to assess landscape diversity. The integration of our work with the collection of recent large-scale surveys is leading to a synoptic view of protist ecology (23, 24, 28, 42, 43, 54–57).

**Surface water assemblages contain high proportions of phototrophs and heterotrophs.** Lacustrine protist diversity was dominated by ochrophytes, which accounted for both the highest sequence abundance and ASV richness. Cryptophytes, ciliates, dinoflagellates, and chlorophytes were also highly represented, reflecting taxonomic profiles typical of freshwater biomes on other continents and identified with primer pairs targeting other gene regions (58). The main taxa contributing to the dissimilarity between assemblages were mixotrophic cryptophytes, whose distributions as major bacterivores may be dependent on the occurrence of specific prey (59).

Phototrophic taxa accounted for the greatest richness and relative abundance of ASVs,

**FIG 6** Legend (Continued)
are shown for the top 7 ASVs contributing to the variation explained by the first two PC dimensions. Ecozone affiliations and trophic state classifications of lakes are represented by letter symbols and colour, respectively.

perhaps not unexpectedly given that assemblages were sampled from the euphotic zone. Notably, the dominance of phototrophs within the protist fraction appears to be a distinct feature of freshwater photic zones in contrast with the prevalence of heterotrophs in the sunlit ocean and surface soils (13). We showed that heterotrophs and mixotrophs were numerically important groups after phototrophs, which, along with a smaller collection of parasites, illustrates that lake surface waters harbour a broad array of microbial functions linking multiple trophic levels.

**Protist communities respond to local environmental conditions.** The key environmental drivers of protist diversity in Canadian lakes differed from those observed in previous large-scale surveys. Assemblages showed the strongest responses to physicochemical factors, including nutrient, major ion, and chlorophyll *a* concentrations, pH, and lake colour. The environmental drivers in lakes differed from those in marine and soil ecosystems, where protist diversity on a large scale is generally predicted by temperature (60) and annual precipitation (28), respectively. The differences in environmental filtering between biomes likely represent fundamental differences in the types of habitats and degrees of ecosystem connectivity. Compared with the more spatially continuous and expansive ocean and soil macroenvironments, lakes are fragmented across the landscape, and neighbouring lakes can exhibit widely contrasting physicochemical attributes (10, 61). Lake heterogeneity is amplified by temporal variability and punctuated perturbations. Higher community turnover has previously been measured among lakes than within marine or soil ecosystems (12, 13), which we reassert is linked to physicochemical heterogeneity.

We found that the influence of local environmental conditions far outweighed the effects of geographic variation across the continental extent. Among LakePulse sites, systems with the highest trophic states (typically, Prairies and Boreal Plains lakes) were located at intermediate longitudes, latitudes, and altitudes. A study of Scandinavian boreal lakes spanning a longitudinally aligned and narrower trophic state gradient (oligotrophic to mesoeutrophic) reported that geography explained more protist community variation than water chemistry (42), complementing our assessment that regional physicochemical heterogeneity is a major determinant of protist diversity. Other surveys identified biogeographic patterns of protist diversity structured by the isolation and dispersal limitation of mountain lake communities (41, 43). Geographic barriers (e.g., the Rocky Mountains) did not appear to generate strong compositional divisions in our set of protist assemblages but were identified as having an important influence on the distributions across LakePulse sites of crustacean zooplankton (62), a group with greater dispersal limitation due to their larger body sizes. Instead, we found the greatest taxonomic and phylogenetic divisions between regions distinguished by differences in lake trophic state and other local environmental conditions.

Partitioning protist trophic diversity allowed us to examine how different components of freshwater food webs respond to the environment. We observed contrasting distribution patterns for each trophic mode. Phototroph assemblages among hypereutrophic lakes exhibited significantly lower taxonomic turnover (as evaluated by mean distance to the trophic state median) than lakes at lower nutrient states, whereas heterotroph assemblages did not follow this trend but turned over significantly more rapidly among eutrophic lakes than mesotrophic or oligotrophic lakes. However, the turnover within each trophic mode was predicted by a mostly overlapping suite of physicochemical factors, including chlorophyll *a*, DIC, ions, pH, and colour, although rank order of importance varied among groups. Nutrient (TP and TN) concentrations and surface temperature were exclusive predictors of heterotroph turnover. All of the environmental predictors of mixotroph turnover were common to both phototrophs and heterotrophs, with no predictors unique to mixotrophs. Changes in low levels of chlorophyll *a* were associated with the most rapid turnover in phototroph composition, which is to be expected given that chlorophyll *a* is linked to phytoplankton biomass and phototroph diversity shifts along a lake productivity gradient. The next strongest predictor of phototroph turnover was potassium, which is not a limiting resource (63) but displayed extreme regional variation, peaking in

the Prairies, followed by the Boreal Plains. Heterotroph and mixotroph turnover was primarily predicted by DIC concentrations. Overall, the environmental drivers of heterotroph diversity mostly reflect the bottom-up controls on primary producers traversing multiple trophic levels but are rendered more complex by the added effects of nutrient and temperature factors. Bacterial prey and top-down controls, encompassing predation and parasitism (not measured in this study), likely also determine trophic functional diversity.

Mixotroph distributions were the least aligned with trophic state and the least predictable from environmental conditions, which should be expected for organisms with the metabolic versatility to occupy variable niche spaces. The balance between primary production and prey consumption is dependent on a mixotroph's phenotypic plasticity (64). While mixotrophy is competitively advantageous over obligate phototrophy or heterotrophy under low-nutrient conditions (65), primarily phototrophic mixotrophs prevail in oligotrophic lakes and are replaced by primarily heterotrophic mixotrophs as trophic state increases (66). Another variable driving mixotroph diversity in aquatic ecosystems is light availability (67, 68), which is modulated by lake colour, a predictor identified in this study. Here, too, bacterial prey and zooplankton predators with a preference for nutritious mixotrophs likely also exert controls (69).

**Hypereutrophic lakes are taxonomically distinct.** Hypereutrophic lakes, located mostly in agricultural watersheds, contained protist assemblages with the lowest diversity and highest taxonomic distinctness (i.e., LCBD) relative to other lakes in the landscape. Specifically, Shannon diversity was inversely related to ion and nutrient concentrations, while taxonomic distinctness tracked with ion- and nutrient-rich conditions. Hypereutrophic conditions potentially filter protist communities by creating relatively extreme conditions (e.g., light attenuation), tolerated by a small number of taxa assembling into uneven communities distinct from those in lakes at lower-nutrient states. Phosphorus is often the limiting nutrient of phytoplankton in numerous freshwater systems (70, 71), yet TP was a predictor of turnover exclusive to heterotrophs. Phosphorus was found to be an important predictor of littoral protist diversity in European lakes (44) and of long-term microeukaryote community turnover evidenced from paleolimnological trends (19). Interactions between protists and *Cyanobacteria* likely also play a role in determining protist assemblages in hypereutrophic lakes, especially as *Cyanobacteria* of the genus *Microcystis* were found to be associated with high-nutrient conditions across LakePulse sites (72).

Ordinations of separate trophic modes showed that the high community variation among eutrophic lakes was generated by heterotroph diversity, whereas obligate phototroph assemblages were the least varied under hypereutrophic conditions. A positive relationship between compositional heterogeneity and trophic state runs counter to the expectation of reduced community variability (e.g., for phytoplankton [73]) that is predicted to follow the leveling of abiotic conditions among lakes induced by land use and eutrophication. Biotic homogenization appears to have trended with long-term climate warming and eutrophication in *Cyanobacteria* (74) and protist (20) assemblages reconstructed from sediment core chronologies. Following our observation of increased compositional heterogeneity among lakes as a function of trophic state, we posit that protist communities in productive lakes are less stable over time, including over the same season, as observed in bacterial time series (75). The temporal fluxes in abiotic conditions prompting succession may be induced by allochthonous inputs or nutrient resuspension from sediments accrued at higher rates in regions of extensive lake use and high populations densities (76). Furthermore, given that many Prairies and Boreal Plains lakes are shallow and exposed to winds, temporary stratification followed by destratification is not uncommon (77). We speculate that taxon replacement linked to land use and eutrophication may force a re-evaluation of human impact on biodiversity. In particular, anthropogenic pressures may not inherently decrease diversity but instead increase turnover, possibly at the expense of rare or specialist taxa disappearing from the landscape pool (78).

**Watersheds influence lacustrine protist diversity.** While the importance of physicochemical factors for lacustrine protist diversity has been described (42, 44) and elaborated upon in this study, the influence of the watershed, in particular soil properties

and land use, which are often but not entirely correlated with lake physicochemical attributes, until now has not been documented on a continental scale. We found that the taxonomic distinctness of local assemblages (i.e., LCBD) corresponded strongly with the proportion of crop agriculture in the watershed, while surface soil pH was an important predictor of community turnover. Because of their concave topographies and position in the landscape, lakes are recipients of major allochthonous subsidies, with global effects (e.g., carbon storage) that are disproportionate to the spatial extent of lakes (6). The influence of terrestrial catchments on within-lake community dynamics is compelling evidence for why lakes cannot be studied apart from their watersheds.

The filtering of lacustrine protist diversity by watershed soil chemistry points to the influence of external factors on lake conditions. Soil buffering capacity, determined by soil texture, organic matter content, and mineral composition, is a main abiotic control on lake water pH (79). Furthermore, soil properties control the mobility of nutrients and their eventual input into lakes. Specifically, soil pH determines the availability and chemical forms of nutrients, and particle size governs the movement of groundwater carrying released nutrients (80). Soil properties also determine the composition and activity of soil microbiomes, which perform biogeochemical transformations modulating the availability of nutrients (81) and seed potential colonists from soils to lakes. Altered precipitation regimes and warming temperatures associated with climate change are expected to increase soil erosion (82) as well as terrestrial nutrient exports (83).

Given the continual increase in land surface transformed by agricultural production and urbanization (84), accelerated watershed land use conversions are widespread. Changes in soils associated with human activities range from increased nutrient loading and acidification by nitrogenous fertilizers in agriculture (85) to shifts in carbon storage precipitated by changes in land management practices or climate (86). Moreover, land use and climate change interact to increase the frequency and magnitude of nutrient and carbon pulses to water bodies (87). In the interest of securing a healthy future for critical freshwaters, we suggest that current soil chemistry heterogeneity across the landscape can inform predictions about the potential consequences of anthropogenic watershed alterations for lakes. In particular, work can be done to understand the microbial diversity and food web dynamics driven by various soil states in future land use scenarios. Overall, the ability to predict lacustrine protist diversity from watershed conditions, as demonstrated in this study, highlights an expanded potential for monitoring lake ecosystems using remote sensing products (88).

**Conclusion.** This is the first study to examine the taxonomic and trophic functional variation in protist diversity across the expansive and lake-rich Canadian landscape. We showed that on this continental scale, lakes displayed broad environmental heterogeneity, including substantial variation in local physicochemical conditions driving taxonomic and phylogenetic community turnover. Watershed soil pH and crop agriculture additionally predicted community turnover and exceptional local-scale diversity. Hypereutrophic lakes were found to contain less diverse and more distinct assemblages than lower-nutrient lakes, primarily as a product of their variable heterotroph and mixotroph compositions. In contrast, phototroph assemblages were more similar among hypereutrophic lakes. While phototrophy was the prevailing nutritional strategy in lake euphotic zones, heterotrophy was nearly as numerically important; each of these trophic modes was highly predictable from physicochemical and other environmental factors. Our survey and findings serve as a valuable resource for mapping species distributions and provide a basis for future research into the increasing anthropogenic impact on lake microbiomes.

## MATERIALS AND METHODS

**Lake selection and sampling.** Hundreds of lakes were sampled between July and early September in 2017 to 2019 by the Natural Sciences and Engineering Research Council of Canada (NSERC) Canadian Lake Pulse Network (52). Sampling was timed to coincide with the summertime period of water column thermal stratification, where relevant. Lakes were sampled across 12 terrestrial ecozones, regions defined by landform, geology, and vegetation (89). Lake selection was stratified across lake surface area and watershed land use impact categories to capture natural and human-mediated lake heterogeneity. Only

natural lakes with a maximum depth of at least 1 m and within 1 km from a road were considered. Freshwater to oligosaline lakes (identified as having conductivity of <8 mS/cm and total major ion concentrations of <4,000 mg/L) were selected for this analysis.

Water was collected using an integrated tube sampler from the euphotic zone over a depth of up to 2 m below the surface at the deepest point in the lake (90). The site of maximum lake depth was located by depth sounding with the aid of bathymetric maps where available. The depth of the euphotic zone was estimated as twice the Secchi disk depth. All water sampling equipment was acid washed and rinsed three times with lake water before use. Carboys were stored in icepack-chilled coolers until water filtration later in the day. Water was prefiltered through 100-$\mu$m nylon mesh and vacuum filtered on 47-mm-diameter 0.22-$\mu$m Durapore membranes through a glass funnel at a maximum pressure of 8 inHg. Up to 500 mL of water was filtered until the filter was nearly clogged. Filters were stored in sterile cryovials at $-80°$C.

**18S rRNA gene amplification and sequencing.** DNA was extracted using the DNeasy PowerWater kit (Qiagen, Hilden, Germany) according to the manufacturer's instructions with the addition of two optional steps: after bead beating and centrifugation, 1 $\mu$L RNase A was added to samples, followed by 30 min incubation at 37°C. DNA was quantified using the Qubit dsDNA BR assay (Invitrogen, Carlsbad, CA, USA). A $\sim$265-bp fragment of the 18S rRNA gene V7 region was amplified with the primers 960F (5′-GGCTTAATTTGACTCAACRCG-3′) (91) and NSR1438 (5′-GGGCATCACAGACCTGTTAT-3′) (92) to broadly target microeukaryotes (93). Each PCR contained a total 25-$\mu$L mixture of 14.25 $\mu$L Milli-Q water, 5 $\mu$L 5$\times$ High-Fidelity buffer, 1.25 $\mu$L of each 10 $\mu$M primer, 0.5 $\mu$L 10 mM deoxynucleoside triphosphates (dNTPs), 0.25 $\mu$L dimethyl sulfoxide (DMSO), 0.5 $\mu$L Phusion DNA polymerase (Thermo Fisher Scientific, Waltham, MA, USA), and 2 ng DNA template. PCR conditions consisted of an initial denaturation at 98°C for 1 min, 30 cycles of 98°C for 10 s, 60°C melting for 30 s, and 72°C for 20 s, and a final extension at 72°C for 5 min. PCR products were loaded with Orange G dye on an ethidium bromide-stained 2% agarose gel and electrophoresed at 40 V for 100 min. DNA bands aligned at the target fragment length against a 100-bp DNA ladder were excised with razor blades and gel extracted with the QIAquick gel extraction kit (Qiagen, Hilden, Germany), modified by final elution into Milli-Q water. PCR products were submitted to Genome Quebec for library barcoding and sequencing of 250-bp paired-end reads in three sequencing runs on an Illumina MiSeq platform.

**ASV inference and annotation.** Primer sequences were removed in Cutadapt v. 3.1 (94). Trimmed reads were processed into ASVs through DADA2 v. 1.16 (95). Samples were pooled for ASV inference using otherwise default parameters. Taxonomy was assigned with naive Bayesian classification trained on PR$^2$ v. 4.12.0 (96). Potentially spurious ASVs were removed by visually inspecting a *de novo* alignment performed in MAFFT (97). To retain only putative protist ASVs, ASVs assigned to Metazoa, Fungi, and Embryophyceae were removed. Taxa were assigned to trophic functional groups as either photoautotrophs, heterotrophs (bacterivores, cytotrophs, saprotrophs, or osmotrophs), mixotrophs, or parasites according to lineage-specific feeding habits summarized by Adl et al. (98).

**Sequence similarities with known diversity.** A database of 18S rRNA gene references restricted to the V7 region was constructed by applying the 960F/NSR1438 primer pair to PR$^2$ v. 4.13.0 in Cutadapt. ASV top hits were queried against the PR$^2$ database in BLAST v. 2.6.0+ (99). Sequence identities were reported for ASVs that were globally aligned to references over a length $\geq$220 nucleotides.

**Environmental data collection.** Lake trophic states were assigned based on TP concentration thresholds estimated for Canadian freshwater systems: ultraoligotrophic (TP concentration, <4 $\mu$g/L), oligotrophic (4 to 10 $\mu$g/L), mesotrophic (10 to 20 $\mu$g/L), mesoeutrophic (20 to 35 $\mu$g/L), eutrophic (35 to 100 $\mu$g/L), and hypereutrophic (>100 $\mu$g/L) (100). Meteorological conditions recorded over 7 days leading up to sampling and ice disappearance day data were accessed from ERA5-Land hourly reanalysis (101). Data on watershed slope and lake volume, discharge, and hydraulic residence time were accessed from HydroLAKES v. 1.0 (5). Watershed surface soil properties were accessed from SoilGrids250m (102). Land cover information was compiled as described by Huot et al. (52). Maps were constructed in R with the NAD 83 coordinate reference system and using the coordinates of Canada from the package maps (103) and ecozone shapefiles sourced from the Canada Council of Ecological Areas (89).

Environmental data were categorized into thematic groups of variables. Latitude, longitude, and altitude were categorized as geography variables. Ice disappearance day and meteorological variables (air temperature, precipitations, and net solar radiation) were categorized as weather variables. Lake surface area, circularity, volume, maximum depth, discharge, residence time, watershed slope within 100 m of the shoreline, watershed area, and lake-to-watershed area ratio were categorized as lake morphometry variables. Watershed land use (crop agriculture, pasture, built development, and clear-cut forestry) and natural land cover fractions, human population density, and mean surface soil properties (bulk density of the fine earth fraction, cation exchange capacity, nitrogen, pH, organic carbon density, organic carbon content in the fine earth fraction, volumetric fraction of coarse fragments, clay, sand, and silt) were categorized as watershed variables. Surface water temperature, calcium, magnesium, potassium, sodium, chloride, sulfate, TP, TN, DIC, DOC, and chlorophyll *a* concentrations, pH, and lake colour were categorized as lake physicochemical variables. Missing physicochemical data were replaced with ecozone median values. Highly colinear variables, evaluated by a Pearson's correlation *r* value of $\geq$0.7, within all categories except physicochemistry were removed.

**Diversity analyses.** ASVs were aligned in the SILVA Incremental Aligner v. 1.7.2 (104) against the SILVA 138.1 SSU Ref NR 99 database (August 27, 2020 release) (105). A maximum-likelihood phylogeny was constructed in FastTree v. 2.1.11 using the Generalized Time-Reversible model of nucleotide evolution (106). Phylogenetic dissimilarities between ASV assemblages were calculated as generalized UniFrac distances, which are sensitive to compositional changes in lineages of intermediate abundance (107). Generalized UniFrac distances ($\alpha = 0.5$) were computed using the GUniFrac package.

To deal with uneven total sequence abundance across samples, sequence count composition was scaled to relative abundance. Rarefaction analysis was conducted on the total data set by measuring ASV richness in assemblages randomly subsampled at each 1,000-sequence step. Taxon accumulation was estimated in a random ordering of lakes using 100 permutations in the package vegan (108). Local diversity indices (richness, Pielou's evenness, Shannon diversity, and Faith's phylogenetic diversity) were calculated from rarefied community data (i.e., randomly subsampled to the lowest sample abundance equaling 10,069 sequences) in the R packages vegan (108) and picante (109). PCAs were computed on Hellinger-transformed community data in vegan.

LCBD and $\beta$-diversity partitioning analyses were performed on $\beta$-diversity estimated using 100 permutations from Hellinger-transformed community data in the package adespatial (110). RV coefficients were computed from the first two principal components or coordinates in the package FactoMineR (111). Nonlinear relationships between $\beta$-diversity and untransformed environmental gradients were modeled in GDMs (112, 113) in the package gdm (114). To create GDM site-pair tables, pairwise dissimilarities between sites were weighted proportionally to the total number of sequences associated with each sample. Variable selection for GDMs was performed using backward elimination with 100 permutations per step. To assess the dispersion in taxonomic composition among lakes of the same trophic state, the mean Bray-Curtis distances of assemblages to trophic-state medians (i.e., centroids) calculated across the first two principal coordinates were compared using ANOVAs, followed by *post hoc* Tukey's tests.

Data wrangling and statistical analysis were performed in R v. 4.0.2 (115).

**Data availability.** Sequence data have been deposited in the European Nucleotide Archive under study accession number PRJEB42538 (www.ebi.ac.uk). Scripts are accessible from https://github.com/rebeccagarner/lakepulse_protists.

## SUPPLEMENTAL MATERIAL

Supplemental material is available online only.
**FIG S1**, PDF file, 0.8 MB.
**FIG S2**, PDF file, 0.8 MB.
**FIG S3**, PDF file, 0.4 MB.
**FIG S4**, PDF file, 0.4 MB.
**FIG S5**, PDF file, 1.4 MB.
**FIG S6**, PDF file, 7.6 MB.
**FIG S7**, PDF file, 1.5 MB.
**FIG S8**, PDF file, 10.1 MB.
**TABLE S1**, CSV file, 3.9 MB.
**TABLE S2**, DOCX file, 0.02 MB.

## ACKNOWLEDGMENTS

This research was funded by the NSERC Canadian Lake Pulse Network (Strategic Network Grant NETGP-479720) and Canada Research Chairs held by D.A.W., I.G.-E., and Y.H. R.E.G. and V.E.O. were supported by the NSERC CREATE ÉcoLac training program in lake and fluvial ecology. R.E.G. also acknowledges support from a *Fonds de recherche du Québec—Nature et technologies* doctoral research scholarship and the Stephen Bronfman Scholarship in Environmental Studies.

We thank the LakePulse sampling crews and the many landowners, municipal and park employees, lake associations, and First Nations who welcomed and facilitated the sampling effort. We also thank the researchers in LakePulse who contributed to the data set, Zofia Taranu for helpful discussions on statistical methods, and Paul MacKeigan, Cindy Paquette, Marieke Beaulieu, and Katherine Griffiths for their collaboration on data quality assessment and curation.

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
