## [Reviewer comments · mSystems]

Protist diversity and metabolic strategy in freshwater lakes are shaped by trophic state and watershed land use at a continental scale

Rebecca Garner, Susanne Kraemer, Vera Onana, Yannick Huot, Irene Gregory-Eaves, and David Walsh

Corresponding Author(s): Rebecca Garner, Concordia University

Review Timeline:

Submission Date:

April 1, 2022

Accepted:

May 13, 2022

Editor: Ashley Shade

Reviewer(s): The reviewers have opted to remain anonymous.

Transaction Report:

DOI: <https://doi.org/10.1128/msystems.00316-22>

May 13, 2022

Mx. Rebecca Garner
Concordia University
Biology
7141 Sherbrooke St. West
L-GE 300.03
Montreal, Quebec H4B 1R6
Canada

Re: mSystems00316-22 (Protist diversity and metabolic strategy in freshwater lakes are shaped by trophic state and watershed land use at a continental scale)

Dear Mx. Rebecca Garner:

My apologies for the delay in decision, which was because of reviewer availability.

Your manuscript has been accepted, and I am forwarding it to the ASM Journals Department for publication. For your reference, ASM Journals' address is given below. Before it can be scheduled for publication, your manuscript will be checked by the mSystems production staff to make sure that all elements meet the technical requirements for publication. They will contact you if anything needs to be revised before copyediting and production can begin. Otherwise, you will be notified when your proofs are ready to be viewed.

Publication Fees:

We recognize that the video files can become quite large, and so to avoid quality loss ASM suggests sending the video file via <https://www.wetransfer.com/>. When you have a final version of the video and the still ready to share, please send it to mSystems staff at mssystems@asmusa.org.

For mSystems research articles, if you would like to submit an image for consideration as the Featured Image for an issue, please contact mSystems staff at mssystems@asmusa.org.

Sincerely,

Ashley Shade
Editor, mSystems

Journals Department
Figure S6: Accept
Figure S8: Accept
Table S2: Accept
Figure S3: Accept
Figure S4: Accept
Figure S5: Accept
Figure S7: Accept
Figure S2: Accept
Figure S1: Accept
Table S1: Accept